# Analysis of Spatiotemporal Changes of Ecological Environment Quality and Its Coupling Coordination with Urbanization in the Yangtze River Delta Urban Agglomeration, China

**DOI:** 10.3390/ijerph20021627

**Published:** 2023-01-16

**Authors:** Zhiyu Shi, Yating Wang, Qing Zhao

**Affiliations:** School of Geography, Geomatics and Planning, Jiangsu Normal University, Xuzhou 221116, China

**Keywords:** eco-environment quality, Google Earth Engine, spatiotemporal variation, MRSEI, urbanization, the Yangtze River Delta urban agglomeration

## Abstract

It is inevitable that urban agglomeration will have a coercive impact on the regional Ecological Environment Quality (EEQ) as a consequence of high-speed urbanization. Balancing the EEQ and urbanization development has become a problem worthy of attention. In order to objectively evaluate the EEQ of the Yangtze River Delta Urban Agglomeration (YRDUA) and explore the impact of the urbanization process on it, this paper is based on the Modified Remote Sensing Ecological Index (MRSEI) and the Comprehensive Night Light Index (CNLI), respectively, and conducts a quantitative assessment of the YRDUA in China from 2000 to 2020. The results show that: (1) From 2000 to 2020, the MRSEI of the YRDUA first decreased and then increased, and the ecological environment quality degraded first and then improved; however, there were significant differences between regions. The ecological environment quality in the south is obviously better than that in the north, and the ecological environment quality in the north changes more drastically, and the low value area of MRSEI will gradually move downstream as time changes; (2) During the study period, the YRDUA formed a hierarchical and progressive urbanization pattern. The inland urbanization process expanded from east to west along the Yangtze River, and the urbanization process of coastal cities expanded from Shanghai as the center to the north and south with high-intensity urbanization cities concentrated in Shanghai and its surrounding cities and low-intensity urbanization cities distributed in the western part of the urban agglomeration; (3) The Coupling Coordination Degree (CCD) between urbanization and EEQ in the YRDUA has continuously improved with an increase of 28.57% in the past 21 years, and the number of cities with high level coupling continues to rise, while the number of medium level coupling cities and low level coupling cities has declined. As a large-scale and long-term analysis of changes in the EEQ and the urbanization process, this study can provide theoretical support for policymakers to formulate mesoscale development planning, EEQ monitoring, and environmental protection policies.

## 1. Introduction

At present, urbanization is a powerful driving force for sustainable and healthy economic development which has greatly promoted regional economic development. China is considered to be a developing country with the fastest urbanization rate and a high urban population which has experienced rapid urbanization and energy consumption [1,2]. However, urban sprawl, irrational land development and utilization in the process of rapid urbanization have caused various ecological problems including severe air pollution, habitat fragmentation, and ecosystem degradation which limit urbanization’s sustainable development [3,4,5]. With the advancement of urbanization and the development of the regional economy, there is an increasingly evident trend in regional agglomeration in China’s urban development. Based on one or more central urban areas, urban agglomeration develops through edge, axial or multi center expansion, and finally forms regional integrated development. Urban agglomeration has become an essential platform for promoting China’s economic growth and participation in international competition and cooperation [6,7]. With the advancement of urbanization, the intensity of human activities has increased, and the consumption of natural resources and energy has become more concentrated; therefore, large urban agglomeration face increasingly severe pressure in Ecological Environment Quality (EEQ). Two aspects deserve attention during the interaction between urbanization and the EEQ. Rapid urbanization will inevitably lead to regional EEQ degradation, while the deterioration of EEQ will restrict urbanization and sustainable development. As the contradiction between urbanization and the EEQ in regional development becomes more and more prominent, effective models are required to identify the spatiotemporal changes in regional EEQ, to improve the EEQ of large urban agglomerations and achieve the coordinated development of the economy and environment.

Recently, remote sensing technology has advanced rapidly, thereby providing data sources and technological assistance for regional EEQ monitoring and evaluation which can accurately reflect ecological conditions across scales [8,9,10]. Various ecological indicators based on remote sensing have played an important role in reflecting and quantifying the quality and function of the ecological environment [11]. Xu [12] proposed to acquire the following four evaluation indexes from remote sensing images: the Normalized Differential Vegetation Index (NDVI), the Wetness(WET), the Land Surface Temperature(LST) and the Normalized Difference Build-up and Soil Index(NDBSI). Then, based on the four ecological indexes, the Remote Sensing Ecological Index (RSEI) was constructed using Principal Component Analysis (PCA) to quantitatively evaluate the regional EEQ. RSEI can integrate multiple ecological indicators to objectively and quickly evaluate regional EEQ. Moreover, as the above four factors can apply the same remote sensing data source, the change or error of weight definition caused by individual characteristics can be avoided when using RSEI to evaluate EEQ. The reliability and applicability of RSEI have been verified in numerous previous studies [10,13,14]. However, RSEI emphasizes near-surface elements including vegetation, humidity, hotness and dryness. The ecosystem is a complex and diversified system, especially in large urban agglomerations with a high degree of industrialization and urbanization. There are many factors that affect the internal EEQ, among which air quality significantly influences the EEQ of urban agglomeration. In recent years, with the rapid development of industrial cities, the emission of atmospheric pollutants including industrial coal consumption, automobile exhaust emissions and dust emissions from construction sites have increased sharply, leading to the continuously intensified urban air pollution in China, especially in areas with better economic development, such as the Yangtze River Delta, the Pearl River Delta and the Beijing-Tianjin-Hebei region [15,16]. A large number of atmospheric pollutants were discharged into the atmosphere, causing increasingly severe air pollution [16,17]. A high concentration of inhalable atmospheric particulate matter PM2.5 (particulate matter with airborne diameter ≤ 2.5 μm) is the main reason for urban air pollution [17,18]. Aerosol Optical Depth (AOD) is an important index to evaluate the degree of atmospheric pollution which can be applied to study the spatial distribution of global and regional PM2.5 concentrations [19]. At present, AOD products based on moderate Resolution Imaging Spectroradiometer (MODIS) have been verified globally and can well reflect the spatiotemporal distribution and concentration changes of PM2.5 [20,21]. In addition, the calculation of RSEI based on traditional professional remote sensing software is very complex and inefficient. Google Earth Engine (GEE) can obtain a large amount of image data online and process the image online, which greatly improves the efficiency of work. It has leading advantages in remote sensing application research of large-scale and long- time series [22,23]. The EEQ monitoring and evaluation research based on GEE has been applied in numerous academic research settings, domestically and abroad [24,25].

Nighttime lighting (NTL) data can reflect human activities and economic development, and is comprehensively applied to socio-economic parameter estimation [26,27,28], population distribution estimation [29,30], energy and power consumption [31,32,33], urbanization monitoring [34,35,36], and urban agglomeration evolution [37,38] which has become a major source for monitoring human activities and socio-economic development. Numerous studies adopted nighttime light data to extract urban scope and measure urbanization level, and their results show that nighttime light data can characterize urbanization and urban expansion [39,40]. Supported by the rapid development of sensor technology in recent years, nighttime light data has been upgraded from the initial low-resolution Defense Meteorological Satellite Program/Operational Linescan System(DMSP/OLS) data to higher-resolution National Polar-Orbiting Partnership/Visible Infrared Imaging Radio-meter Suite(NPP/VIIRS) data with more robust nighttime light perception capability that is more competent in efficiently extracting Urban Built-Up Areas (UBA) and effectively monitoring urban spatiotemporal changes.

The Yangtze River Delta Urban Agglomeration (YRDUA) is one of the most developed regions in China and has one of the highest urbanization rates [41]. With the continuous improvement of the economy and urbanization in the YRDUA, the pressure on the regional ecological environment is becoming increasingly severe. Assessing and analyzing the changes of EEQ in the YRDUA and its coupling with the urbanization process is of great significance for achieving the coordinated development of economy and environment. Consequently, this paper employed the PCA to couple the greenness, wetness, heat, dryness, and air quality to construct the Modified Remote Sensing Ecological Index (MRSEI), which was used to assess and analyze the EEQ of the YRDUA. Then, the comprehensive nighttime light index (CNLI) was calculated based on NTL to evaluate the urbanization level of the YRDUA. Finally, CNLI and MRSEI were used to construct the coupling coordination model of urbanization and EEQ, and the interaction between EEQ and the urbanization process in YRDUA were explored in this paper. The specific objectives of this study are: (1) exploring the spatiotemporal changes of EEQ in the YRDUA under the background of rapid urbanization; (2) evaluating the spatiotemporal changes of urbanization level in the YRDUA from 2000 to 2020; and (3) evaluating the changes of coupling coordination level between the EEQ and the urbanization level in the YRDUA. The research results are expected to promote sustainable development in the YRDUA and provide a reference for high-quality construction of urban agglomeration in China.

## 2. Study Area and Data

### 2.1. Study Area

The YRDUA is located in the lower reaches of the Yangtze River adjacent to the Yellow Sea and the East China Sea. The Yangtze River Delta includes 26 cities including Shanghai, Nanjing, Hangzhou, Hefei, etc., with an area of about 210,000 km^2^. The YRDUA belongs to the subtropical monsoon climate. Most of the terrain is plain and the terrain is high in the south and low in the north [42]. The study area is shown in Figure 1. In 2020, the YRDUA accounted for approximately 11% of the country’s total population and 19% of the country’s total GDP, thus making it one of the regions with the fastest economic growth, the highest level of urbanization, and the largest population absorption in China.

### 2.2. Data and Pre-Processing

#### 2.2.1. MODIS Data

Five ecological indexes were used to construct the MRSEI in this research, including greenness, wetness, heat, dryness, and air quality. The MODIS is commonly used in EEQ research because of its moderate spatial resolution, high image quality, and short monitoring period. All MRSEI indicators were computed using MODIS data products from 2000 to 2020 because of the broad area and cloud cover of remote sensing pictures in the research region. Specific data are shown in Table 1.

Based on the GEE, this paper selected the high-quality images online with few clouds from May to October 2000–2020. Then the remote sensing image was pre-processed to remove cloud pixels and water mask, and each type of remote sensing data was multiplied by the corresponding conversion coefficient. In addition, all types of MODIS data products were unified and resampled to 500 m × 500 m. The data acquisition and processing address is: https://code.earthengine.google.com/4f607932c615becde3815e591ad62996 (accessed on 5 November 2022).

#### 2.2.2. Nighttime Remote Sensing Data

DMSP/OLS image is released by the National Geophysical Data Center (NGDC) of the National Oceanic and Atmospheric Administration (NOAA), with the download address of https://www.ngdc.noaa.gov/ngdc.html (accessed on 6 November 2022). The current image data sets include 33 phases of images obtained from 6 different DMSP satellites including F10 (1992–1994), F12 (1994–1999), F14 (1997–2003), F15 (2000–2007), F16 (2004–2009), and F18 (2010–2012) for a total of 21 years from 1992 to 2012. NPP/VIIRS image is derived from the Visible Infrared Imaging Radiometer Suite (VIIRS) sensor on Suomi NPP (National Polar-Orbiting Partnership) satellite and is managed and downloaded by the Earth Observation Group (EOG) affiliated with the National Geophysical Data Center (NGDC), with the download address of https://eogdata.mines.edu/products/vnl/ (accessed on 6 November 2022). The detailed parameters of the nighttime remote sensing data utilized in this paper are shown in Table 2. 

This study selected five years of nighttime light data in 2000, 2005, 2010, 2015 and 2020, respectively, to analyze the coupling effect of urbanization and EEQ, of which the nighttime light data in 2000, 2005, and 2010 were from DMSP/OLS Stable Light (STL), with a spatial resolution of about 1 km (the original resolution of DMSP/OLS sensor was 2.8 km, with product resampling of 1 km), where the pixel gray value (DN value) represented the average light intensity, with a range of 0–63. Since there was a light saturation phenomenon with DMSP/OLS data drifting with images from different satellite sensors, this paper adopted a Pixel-Based Pseudo-Invariant Features (PBPIF) based on the fluctuation characteristics of pixels [43] to obtain the calibration light data of 2000, 2005 and 2010 of the study area. The nighttime light remote sensing data in 2015 and 2020 were from NPP/VIIRS Cloud Mask Straight Light (VCMSL). The VCMSL dataset provided the monthly average value of the Day/Night Band, with a spatial resolution of about 500 m. The product corrected the pollution data using the stray light correction method [44], which realized a more refined spatial resolution and removed data saturation. Since the light data in 2015 and 2020 were obtained from NPP/VIIRS and standing at the perspective of the time dimension, they are inconsistent and incomparable with previous lighting data and cannot be directly applied to the study in this paper. This paper applied the calibration model proposed by Li et al. [45] to integrate DMSP/OLS data and NPP/VIIRS data and then applied the low-pass Gaussian filtering to eliminate noise and achieve the matching of light data. In addition, the paper resampled the STL data to a resolution of 500 m and synthesized the mean value of multi-period monthly average VCMSL data to obtain annual data.

## 3. Research Methods

### 3.1. MRSEI Indexes and Calculation

The MRSEI is coupled with five indexes of greenness, wetness, heat, dryness, and air quality, with the calculation formula as follows:(1)MRSEI=f(NDVI,Wet,LST,NDBSI,AOD)
where, NDVI, WET, LST, NDBSI, and AOD represent greenness, wetness, dryness, heat, and air quality, respectively.

#### 3.1.1. Greenness Index

The greenness index represents the coverage and growth of surface vegetation and is essential to measure the regional EEQ. The NDVI is established based on the absorption of green vegetation leaves in the red light band and reflection characteristics of the near-infrared band, which can detect the state and coverage of surface vegetation [46]. Hence, NDVI in the growing season was used to represent greenness index in this study, and the formula is as follows: (2)$$NDVI=(\rho_{\mathrm{nir}}-\rho_{red})/(\rho_{\mathrm{nir}}+\rho_{red})$$
where *ρ_red_* represents the reflectance of the red band, *ρ_nir_* represents the reflectance of the near-infrared band 1(NIR1).

#### 3.1.2. Wetness Index

WET can reflect the humidity of vegetation and soil, which is closely related to the EEQ [47,48]. In this paper, WET was obtained by the tasseled cap transformation based on MOD09A1 [49], and the formula is as follows:(3)$$WET=0.1147\rho_{red}+0.2489\rho_{nir1}+0.2408\rho_{blue}+0.3132\rho_{\mathrm{green}}-0.3112\rho_{nir2}-0.6416\rho_{swir1}-0.5087\rho_{swir2}$$
where *ρred*, *ρnir1*, *ρblue*, *ρgreen*, *ρnir2*, *ρswir1* and *ρswir*2 represent the reflectance of the bands of the red, NIR1, blue, green, near-infrared band 2(NIR2), short-wavelength infrared 1 (SWIR1), and short-wavelength infrared 2(SWIR2) bands of the MODIS image, respectively. 

#### 3.1.3. Heat Index

LST was used to represent the heat index. LST is a crucial parameter that reflects the land surface environment and an integral part of the Earth’s energy balance. The Daily Surface Temperature (DLST) of the MOD11A2 dataset was used to calculate the LST and then convert the grey value to a Celsius temperature. The conversion formula is:(4)LST=0.02DN−273.15
where *DN* is the gray value of MOD13A2 daytime surface temperature band.

#### 3.1.4. Dryness Index

This research selects the Index-based Built-up Index (IBI) [50] and Soil Index (SI) [51] to characterize the land surface dryness jointly. The above two indexes synthesize the NDBSI, and the calculation formula is:(5)$$NDBSI=\frac{SI+IBI}{2}$$
(6)$$SI=\frac{\left[\left(\rho_{swir1}+\rho_{red})-(\rho_{nir1}+\rho_{blue}\right)\right]}{\left[\left(\rho_{swir1}+\rho_{red})+(\rho_{nir1}+\rho_{blue}\right)\right]}$$
(7)$$IBI=\frac{\frac{2\rho_{swir1}}{\rho_{swir1}+\rho_{nir1}}-\left[\frac{\rho_{nir1}}{\rho_{nir1}+\rho_{red}}+\frac{\rho_{\mathrm{green}}}{\rho_{\mathrm{green}}+\rho_{swir1}}\right]}{\frac{2\rho_{swir1}}{\rho_{swir1}+\rho_{nir1}}+\left[\frac{\rho_{nir1}}{\rho_{nir1}+\rho_{red}}+\frac{\rho_{\mathrm{green}}}{\rho_{\mathrm{green}}+\rho_{swir1}}\right]}$$
where *ρ*_1_, *ρ*_2_, *ρ*_3_, *ρ*_4_ and *ρ*_6_ are the *Red*, *NIR1*, Blue, Green and *SWIR*1 bands of the MOD09A1 images, respectively.

#### 3.1.5. Air Quality Index

Air quality is an integral part of the EEQ. Relevant research shows that AOD can accurately reflect the air quality of a particular area [52,53,54]. The study applies MCD19A2 aerosol products to represent air quality. MCD19A2 realizes (MAIAC) algorithm and conducts inversion by applying the multi-angle atmospheric correction. The above algorithm significantly improves the data coverage of dense vegetation areas and bright land surfaces. It also enhances the inversion accuracy by fixing the grid to store the surface spectral and thermal characteristics [54]. Based on the AOD data obtained from Aerosol Robot Network (AERONET), the MAIAC algorithm can realize a higher accuracy than the Dark Target (DT) and Dark Blue (DB) algorithm in Mainland, China [55,56].

#### 3.1.6. Calculation of MRSEI

In order to avoid the impact of the water body on the principal component load and the influence of the non-uniform index dimension on the weight, the following processing method was performed before the PCA: (1) The Modified Normalized Difference Water Index (MDNWI) mask was used to remove water; (2) Since the data dimensions of the five ecological indicators were not uniform, these five indicators were normalized before PCA.

MNDWI calculation formula is as follows:(8)$$MNDWI=(\rho_{\mathrm{green}}-\rho_{\mathrm{mir}})/(\rho_{\mathrm{green}}+\rho_{\mathrm{mir}})$$

The indicator normalization formula is as follows:(9)$$NI=\frac{I-I\min}{I\max-I\min}$$
in which, *NI* represents the normalized index; *I* represents the original ecological index; *I*_min_ is the minimum value of the annual original ecological index, and *I*_max_ is the maximum value of the annual original ecological index.

PCA uses the method of rotating the coordinate axis vertically, in turn, to concentrate the multidimensional information into a few feature components, so as to achieve multi factor dimension reduction. PCA can automatically and objectively determine the corresponding weight of each index according to its contribution to the principal component, which can avoid the result deviation caused by the weight setting. Based on the GEE, five ecological indicators were analyzed by PCA. The first principal component (PC1) contains the maximum information of each variable, which can comprehensively reflect the regional EEQ. Therefore, PC1 is used as the information source of MRSEI_0_. The formula is shown as:(10)$$MRSEI_{0}=PC1\left[f\left(NDVI,WET,LST,NDBSI,AOD\right)\right]$$
where the *PC*1 is the first principal component.

Normalize *MRSEI*_0_, the formula is:(11)$$MRSEI=(MRSEI_{0}-MRSEI_{0-\min})/(MRSEI_{0-\max}-MRSEI_{0-\min})$$
where *MRSEI*_0-max_ represents the maximum value of *MRSEI*_0_, and *MRSEI*_0-min_ represents the minimum value of *MRSEI*_0_. 

### 3.2. Average Correlation Coefficient

The Average Correlation Coefficient(ACC) was used to verify the accuracy of MRSEI representing EEQ in the YRDUA, so as to check the applicability of MRSEI. The ACC refers to the average absolute value of the correlation coefficient of an indicator and other indicators in the same period. If the ACC between *MRSEI* and each index is greater than that between each index, it indicates that MRSEI can represent the EEQ of the YRDUA more comprehensively and accurately than other indexes. The formula of the ACC is:(12)$$\bar{C_{p}}=\frac{\left|C_{q}\right|+\left|C_{r}\right|+\cdots +\left|C_{s}\right|}{n-1}$$
where $\bar{C_{p}}$ represents the ACC; *p*, *q*, *r*, and *s* represent the indexes for correlation analysis, *n* represents the number of indexes for correlation analysis and *C_p_*, *C_q_*, *C_r_*, and *C_s_* is the correlation coefficient among each index.

### 3.3. Construction of the CNLI 

This paper calculated CNLI based on calibrated nighttime light images to reflect the regional urbanization level and surface human activity intensity, which can effectively characterize the development of regional urbanization [10]. The calculation formula is as follows:(13)CNLI=LAP×MLI
(14)$$MLI=\frac{\sum_{i=1}^{63}C_{i}\times DN_{i}}{\sum_{i=1}^{63}C_{i}\times 63}$$
(15)$$LAP=\frac{Area_{light}}{Area}$$
where *CNLI* is the comprehensive nighttime light index, MLI is mean light intensity, *LAP* is the proportion of light area, *DN*_i_ is the gray value of light pixels, and *C_i_* is the number of pixels with *DN*_i_ value; *Area_light_* represents the area of the light patch, and Area represents the total area of the study area.

### 3.4. Coupling Coordination Model

The coupling degree refers to the degree to which two (or more) systems interact with each other through various internal and external factors. The coupling coordination model can reflect the interaction between *MRSEI* and *CNLI*. The coupling degree between EEQ and urbanization development is an important factor affecting the sustainable development of YRDUA. This paper introduces the coupling coordination model of urbanization and EEQ based on the capacity coupling system model in physics [57]. Firstly, the coupling model is:(16)$$CD=2\times \sqrt{(U\times E)/{\left(U+E\right)}^{2}}$$
where *U* is the *CNLI* and *E* is the *MRSEI*; *CD* represents the coupling degree between *CNLI* and *MRSEI*. The larger the *CD* value, the more coordinated the EEQ and urbanization development.

Secondly, in order to avoid “False Coordination” between two systems, this paper introduces a coupling coordination model to objectively reflect the coordinated development level [10], with the formula as follows:(17)$$CCD=\sqrt{(\alpha U+\beta E)\times CD}$$
where *CCD* represents the Coupling Coordination Degree, the higher the *CCD* value is, the higher the coupled and coordinated development level between the two systems. α and *β* are weight coefficients. Since urbanization is the critical factor leading to the EEQ changes, and the impact of the EEQ on urbanization is limited. Therefore, the urbanization system should be given greater weight (*α* = 0.65; *β* = 0.35) [58].

## 4. Results

### 4.1. Analysis of MRSEI Applicability

Table 3 shows that the contribution rates of the PC1 are all over 50%, gathering from most of the information from the five indicators. It indicates that *MRSEI* is capable of representing the regional EEQ, among which, the contribution rates of NDVI and Wet on PC1 are positive, and the contribution rates of LST, NDBSI and AOD are negative, which is consistent with the actual situation. The study calculated the correlation coefficient between *MRSEI*, NDVI, WET, LST, NDBSI and AOD in the same period to test the applicability of the ecology index *MRSEI* (Table 4). Also, it tested the applicability of the model through the mean correlation. During five monitoring years, the mean correlation of *MRSEI* is the largest, ranging from 0.49 to 0.53. Then, this paper shows the calculation of the mean value of the mean correlation over five years. *MRSEI* is still the largest, higher than NDVI, WET, LST, NDBSI and AOD at 0.51, 0.18, 0.22, 0.19, 0.09 and 0.31, respectively. It indicates that *MRSEI* integrates most of the information of each indicator and is more representative than any single indicator. Therefore, *MRSEI* is more competent in comprehensively and widely representing the EEQ of the YRDUA.

### 4.2. Evaluation of EEQ of the YRDUA

#### 4.2.1. Spatiotemporal Pattern of EEQ

Figure 2 shows that the *MRSEI* of the YRDUA fluctuated and increased from 2000 to 2020, with the lowest mean of only 0.537 in 2005 and increased to 0.624 in 2020. The increase of *MRSEI* in the past 20 years shows a specific improvement in the EEQ of the YRDUA. Combining the ecology classification method Hu and Xu [59] provided, *MRSEI* was divided into five grades to better reveal the changes in RSEI in the past 20 years: Extremely Poor (0.0–0.2), Poor (0.2–0.4), Moderate (0.4–0.6), Good (0.6–0.8), and Excellent (0.8–1.0). It can be observed that there is apparent spatial heterogeneity in the distribution of MRESI in the YRDUA. Overall, the south has better EEQ than the north. The regions with “Good” and “Excellent” EEQ grades are primarily concentrated in the south, with a high altitude, low intensity of human activities and high vegetation coverage; the regions with “Moderate” and “Poor” EEQ are mostly distributed in the north and on both banks of the Yangtze River, mainly in urban built-up areas, especially in Hefei and Shanghai. The above regions have a high intensity of human activities, with a high level of urbanization, low vegetation coverage, and poor EEQ. In addition, the EEQ in the northern YRDUA has dramatically changed over the past 20 years, and the low-value area of *MRSEI* has gradually migrated downstream. From 2000 to 2005, there was an extensive deterioration in the EEQ, mainly in the northwest, such as Hefei and Chuzhou, and the east of the YRDUA, such as Shanghai and Suzhou. After 2005, there was an increasing improvement in the overall EEQ of the study area, of which low-value *MRSEI* regions in the north have been vastly reduced, and the low-value *MRSEI* regions are primarily in the urban built-up areas.

The change in area proportion of EEQ at all grades was calculated based on the five *MRSEI* distribution maps in 2000, 2005, 2010, 2015 and 2020 (Figure 3). The results show that in 2000, the EEQ maintained a “Good” grade in Yangtze River Delta, accounting for 58.33%. However, from 2000 to 2005, the area with “Good” EEQ decreased significantly to 26.28%. The regions with “Moderate” and “Poor” EEQ have increased considerably, with area proportion rising from 31.31% and 6.09% to 51.55% and 16.77%, respectively. The EEQ improved continuedly between 2005 and 2020. From 2005 to 2010, the EEQ slightly improved and the EEQ in most areas was Moderate. Since then, the EEQ in the study area improved further. The region of areas with “Good” and “Excellent” EEQ increased continuedly, with the area proportion rising from 30.99% to 54.65%. The change of proportion structure shows that the transformation of *MRSEI* has mainly distributed three grades of “Moderate”, “Good” and “Excellent,” and the change of EEQ is primarily in the “Good” to “Excellent” or “Moderate” grades for the past 21 years.

#### 4.2.2. Spatiotemporal Change Characteristics of EEQ in the YRDUA

To further analyze the change level of EEQ of the YRDUA from 2000 to 2020, this study conducted a pairwise difference analysis on the *MRSEI* index of each year through five grades of *MRSEI*. If the ecology grade rises, the grade difference is positive, which represents the improvement of EEQ in the study area. On the contrary, if the ecology grade decreases, the grade difference is negative, representing that the EEQ in the study area is deteriorating. Accordingly, the paper divides the change amplitude into five grades: “Obvious Improvement (OI),” “Slight Improvement (SI),” “No Change (NC),” “Slight Determination (SD),” and “Obvious Determination (OD)” (Table 5).

In general, the change characteristic of EEQ in most areas of the YRDUA from 2000 to 2020 were NC, accounting for 52.73%, followed by SI and SD, accounting for 25.52% and 21.67%, respectively. The proportion of OD and OI was less than 1%. From 2000 to 2005, the area with deteriorated EEQ in the study area was 86,792.63 km^2^. It was mainly in the SD, accounting for 43.11% of the total area of the study area. From 2000 to 2005, the area with improved EEQ was 14,688.82 km^2^, it was mainly in the SI, accounting for 7.30%, indicating that the overall EEQ of the YRDUA was deteriorated. Then, from 2005 to 2010, the area with deteriorated EEQ in the study area was mainly SD, accounting for 14.94%, and the area with improved EEQ was mainly SI, accounting for 19.94%, with a decreased proportion in ecology deterioration and improvement. From 2005 to 2010, 65.12% of regional EEQ remained unchanged, indicating that the EEQ of the YRDUA was relatively stable. Afterward, from 2010 to 2015, there was 6.16% of the area with deteriorated EEQ and mainly in the SD, and 39.48% of the area with improved EEQ and mainly in the SI, indicating that the EEQ of the YRDUA was improved in a large area from 2010 to 2015. From 2015 to 2020, the area of EEQ deterioration in the YRDUA accounted for 13.89%, and the area of EEQ improvement accounted for 16.91%, indicating that the EEQ change of YRDUA was relatively stable and continued to improve. Therefore, it can be seen that the EEQ of the study area has experienced a process of deterioration first and then improvement afterward.

Figure 4 shows the spatial changes of EEQ. It can be seen that the areas where EEQ has been improved from 2000 to 2020 were mainly concentrated in the southwest and northwest, mainly in Hefei, Chuzhou, Chizhou, Xuancheng and Hangzhou. The regions where EEQ deteriorated were primarily concentrated in the east, mainly in Shanghai, Suzhou, Wuxi, Changzhou, Nantong and Taizhou. From 2000 to 2005, the EEQ in the north and east deteriorated, mainly in Shanghai, Suzhou, Wuxi, Yancheng, Nantong and Jiaxing. Then, from 2005 to 2010, the EEQ of most areas was maintained stably. Ecology deterioration areas were mainly distributed in the south, including Ningbo, Shaoxing and Taizhou. Few ecological improvement areas were distributed northwest, including Hefei and Chuzhou. Afterward, from 2010 to 2015, there was a significant improvement in EEQ, mainly in Yancheng and Nantong in the north and Ningbo, Shaoxing and Taizhou in the south. From 2015 to 2020, the EEQ of most areas remained stable. Ecological improvement areas were mainly distributed in the north, and ecological deterioration areas were primarily distributed in the south.

### 4.3. Nighttime Light in the YRDUA

#### 4.3.1. The Nighttime Light Intensity Distribution and Its Changes

It is widely believed that the nighttime light remote sensing images with necessary preprocessing and calibration can directly or indirectly reflect the scope and intensity of human activities at night. Moreover, it possesses certain advantages in urban issues research including urban expansion and built-up area extraction. Figure 5 shows the calibrated nighttime light images of the YRDUA from 2000 to 2020. Nighttime light areas and high-intensity light pixels significantly increased in the past 21 years. In 2000, most high-brightness pixels were concentrated in major central cities that were independent and disconnected from each other. However, the area of high-brightness pixels has increased dramatically, forming a continuous high-brightness light area. During the research period, there was a relatively low change in light intensity in central urban areas of cities with high urbanization levels (Shanghai, Nanjing, Hefei and Suzhou). On the contrary, there were relatively substantial changes in light intensity in counties and districts around the central urban area. Another noteworthy point was that the nighttime light brightness took Shanghai as the center and extended inland along the Yangtze River during the study period. Meanwhile, the high-brightness nighttime light also expanded to the south and formed a large area of high brightness in the Hangzhou Bay area.

#### 4.3.2. CNLI Changes in Prefecture-Level Cities

This paper calculated the CNLI of each unit and its changes in the scale of prefecture-level cities based on the CNLI, as shown in Figure 6. During the research period, CNLI in the YRDUA area had changed dramatically, indicating significant differences in urbanization levels within the YRDUA. In 2000, only Shanghai’s CNLI value was at a very high level, Wuxi and Suzhou were at a moderate level, and other cities were at a low or very low level, thus indicating that most of the urbanization levels of most cities in the study area were at a low level. In 2005, the CNLI of Suzhou and Wuxi reached the high level. At the same time, the CNLI of Nanjing, Zhenjiang, Changzhou and Jiaxing reached a moderate level. In 2010, the CNLI of Suzhou and Wuxi reached the very high level, and that of Ningbo and Zhoushan reached the moderate level. The high CNLI value of the YRDUA began to show an expansion trend along the Yangtze River and coastline. Since then, compared with 2010, the high value of CNLI further extended “inland” in 2015, when the CNLI of Ma’anshan rose to the low level and Jiaxing rose to the high level. In 2020, the CNLI of Zhoushan rose to the high level. Then, the CNLI of Taizhou, Nantong and Huzhou rose to the moderate level. At the same time, the CNLI of Yancheng, Hefei, Wuhu, Hangzhou, Jinhua and Taizhou rose to the low level. From 2000 to 2020, a well-defined and gradually promoted urbanization pattern in the YRDUA was formed. The inland urbanization process develops from the east to the west along the Yangtze River, while the coastal urbanization process is expanding from Shanghai to the north and south. Moreover, the urbanization cities at a very high level are concentrated in Shanghai and its surrounding cities such as Wuxi, Suzhou and Jiaxing. The urbanization cities at a high level are centered on the areas of high-intensity urbanization. The change gradient of CNLI in different cities shows that the CNLI in eight cities has increased significantly, accounting for 30.77% of all units mainly concentrated in the core areas of the Yangtze River Delta including Nanjing, Zhenjiang, Changzhou, Wuxi, Suzhou, Huzhou and Jiaxing during the 21-year period. These cities have experienced rapid urbanization during the study period and are also one of the most economically dynamic regions in China that shows a high growth rate in CNLI. Eleven cities showed a slightly increased CNLI, that accounted for 42.31% of all cities and were mainly distributed around high-intensity urbanization areas along the Yangtze River to the west, and along the coastline to the north and south. In addition, there are seven cities with slight changes in CNLI that account for 26.92% of all cities. Apart from Shanghai, the initially highly urbanized city, other cities are distributed outside the study area, mainly in the west or north of the Yangtze River Delta, with a relatively low urbanization level and relatively slow urbanization speed.

### 4.4. Interaction between EEQ and Urbanization

Based on the calculated *MRSEI* and urbanization characteristic element CNLI of each city, this paper applies Formulas (16)–(17) to generate the CCD of prefecture-level cities in the YRDUA, as shown in Figure 7. In general, the CCD between *MRSEI* and CNLI shows an upward trend from 2000 to 2020 which indicates that the coupling coordination development of EEQ and urbanization in the Yangtze River Delta had been improved. Specifically, CCD grew at an annual rate of about 0.01 from 0.49 in 2000 to 0.63 in 2020, with a growth rate of 28.57%. The continuous and stable rise of CCD indicates that the EEQ and urbanization of the YRDUA were going through a healthy development process.

To better analyze the coupling coordination characteristics of urban *MRSEI* and CNLI, the CCD of prefecture-level cities was divided into three stages, namely the low-level coupling coordination stage, with CCD lower than 0.45; the middle-level coupling coordination stage, with the CCD between 0.45 and 0.65; the high-level coupling coordination stage, with CCD higher than 0.65, counting the number of cities with different CCD levels, as shown in Figure 8. In 2000, only four cities (15.38%) were in the high-level coupling stage, and 11 cities were in the low-level coupling stage and the middle-level coupling stage, the latter of which took a dominant position. In 2005, the number of cities in the low-level coupling stage maintained stably, which was still at 11. The number of cities in the middle-level coupling stage dropped to eight. Nanjing, Zhenjiang and Changzhou entered a high-level coupling coordination stage. In 2010, the number of low-level coupling cities decreased significantly to six, and the number of medium-level and high-level coupling cities increased, surpassing the low-level coupling cities for the first time. Yancheng, Maanshan, Wuhu, Hangzhou, and Jinhua became medium-level coupling cities. Ningbo and Zhoushan have become high-level coupling cities. In 2015, the proportion of high-level coupling cities increased, reaching 11. The number of cities with medium coupling grades was reduced to nine, and Taizhou and Yangzhou changed from a medium coupling level to a high coupling level. The number of low-level coupling cities was maintained stably at six. In 2020, the number of high-level coupling cities continued to increase, reaching 13 and accounting for 50%. Yangzhou and Huzhou became high-level coupling cities. The number of medium-level and low-level coupling cities decreased, with eight medium-level and five low-level coupling cities. Hefei was changed from a low-level coupling city to a medium-level coupling city.

Reviewing the past 21 years, the CCD of the YRDUA has changed significantly, which was not only represented in area and proportion but also in spatial distribution. In 2000, most of the cities in the study area were low-level coupling cities, while a few high-level coupling cities were clustered around the estuary of the Yangtze River. In 2020, the number of high-level coupling cities reached 50%, and the distribution area expanded significantly. The high coupling level cities were mainly expanded around Shanghai, and the low coupling cities, accounting for 9.23%, were primarily concentrated in the west of the YRDUA. In addition, during the study period, there were two expansion routes for CCD of the YRDUA. One was to expand westward along the Yangtze River, and the other was to expand south-north along the coastal zone with Shanghai as the center, which was basically in line with the expansion trend of CNLI.

## 5. Discussion

As a densely populated and mostly urbanized area, the changes in the EEQ of the YRDUA will have an inestimable influence on the stability and development of the region. Therefore, it is of great significance to comprehensively and objectively realize the monitoring and coupling coordination analysis of the EEQ and urbanization in urban agglomeration for sustainable urban development and green city construction. Meanwhile, considering the characteristics of the EEQ in the study area, the impact of air quality on the EEQ of the study area cannot be ignored. On top of the original RSEI, this paper adds air quality indicators and constructs the *MRSEI* by integrating multi-source remote sensing data and applying the GEE cloud platform to retrieve the temporal and spatial pattern of the EEQ of the YRDUA and analyze the spatiotemporal changes. The results show that *MRSEI* can realize the comparative analysis of regional EEQ and urbanization rapidly and efficiently. It can provide scientific support for the development planning and the monitoring and protection of EEQ in mesoscale regions.

The results of this study are similar to those of previous studies [10,60] which have certain reference value, but there are still some deficiencies in the study. For instance, the *MRSEI* constructed in this paper is completely based on remote sensing images, which means that remote sensing images will also have errors due to different sensors and transit times. In addition, this paper is engaged in ensuring the comparability of *MRSEI* in different years, reducing the image selection time window as much as possible and removing clouds and water from images to avoid interference. However, affected by the quality of MODIS data, the results are still inevitably affected by clouds and other noises, and it is difficult to ensure that the image acquisition time in different years is exactly the same. Therefore, in future research, when calculating *MRSEI*, we should further denoise the image to improve the image quality and ensure the accuracy of the results [61,62,63,64]. Although our research shows the spatiotemporal pattern and changes in EEQ in the YRDUA, the ecosystem is an intricate and varied system. The composition and alterations of EEQ involve many aspects including natural and socio-economic factors. Only five indicators are selected to reflect the EEQ in this study. Future research should consider adopting more refined and diversified index data such as Net Primary Productivity (NPP), Vegetation Health Index (VHI), meteorological drought index, and various socio-economic indicators [65].

Considering the persistence of urban-rural disparity in China, the resource concentration effect of cities will continue to attract people, industries and resources through resource concentration. Since urban expansion is still one of the main trends of the future development of the YRDUA, the coercion effect of urbanization on the EEQ will not disappear in the short term. It is worth noting that based on the coupling coordination model constructed by *MRSEI* and CNLI, this paper can intuitively depict the coupling changes between large urban agglomeration areas and the EEQ during urbanization. However, influenced by the limited spatial resolution of the data and model parameters, it is difficult to further explore the coupling mechanism and driving force between urbanization and the EEQ. Whether the method proposed in this paper can effectively reflect the practical problems also needs to be further tested with non-remote sensing data. In addition, the urbanization grade of different areas within the city varies greatly. The coordination and coupling relationship between the urbanization process and EEQ in urban built-up areas and suburban or rural areas are different.

With the continuous improvement of the quality of the data sources and continuous optimization of the model algorithms, there are several topics that should be focused on in future research such as the introduction of multi-dimensional ecological index parameters (economic, population, and environmental protection policies) to improve the coupling evaluation system of the ecological environment and the urbanization level, the interaction mechanism between urban expansion and the ecological environment in urban development and discussion of the coupling mechanism of urban expansion and ecological environment in different land use types on a smaller spatial scale in order to reveal the relationship between the environment and urban development more comprehensively.

## 6. Conclusions

Combining MODIS images and nighttime light data, this paper constructs *MRSEI* and CNLI to evaluate the ecology status and urbanization intensity of the YRDUA in the past 21 years and explore the coupling and interaction between them.

The results show that: (1) During the research period, the EEQ of the YRDUA deteriorated from 2000 to 2005. Although the EEQ had continuously improved from 2005 to 2020, there were significant differences among regions. The EEQ of the YRDUA in the south was better than that in the north. Most regions with “Good” and “Excellent” EEQ were located in the south of the YRDUA. The areas with “Moderate” and “Poor” EEQ were mostly located in the urban built-up areas in the north of the YRDUA and on both banks of the Yangtze River. In addition, the EEQ in the north of the YRDUA has changed dramatically in the past 21 years, and the *MRSEI* low-value region has shifted downstream. (2) Through 21 years of urbanization, the YRDUA has formed a hierarchical and progressive urbanization pattern, where the inland urbanization process expands from east to west along the Yangtze River. Meanwhile, the urbanization process of coastal cities expanded from Shanghai to the north and south. High-intensity urbanization cities are concentrated in Shanghai and its surrounding cities including Wuxi, Suzhou and Jiaxing. In contrast, low-intensity urbanization cities are distributed in the west of the urban agglomeration. (3) The number of cities with a high-level coupling of EEQ and urbanization in the YRDUA has been increasing continuously in the past 21 years, changing from four to 13. Meanwhile, the number of cities with medium-level and low-level coupling has declined, changing from 11 to eight and five, respectively. The CCD of the YRDUA was increasing at a rate of around 0.01 per year and rose from 0.49 in 2000 to 0.63 in 2020 with an increase of 28.57%. The coupling level between its urbanization and EEQ is constantly improving.

## Figures and Tables

**Figure 1 ijerph-20-01627-f001:**
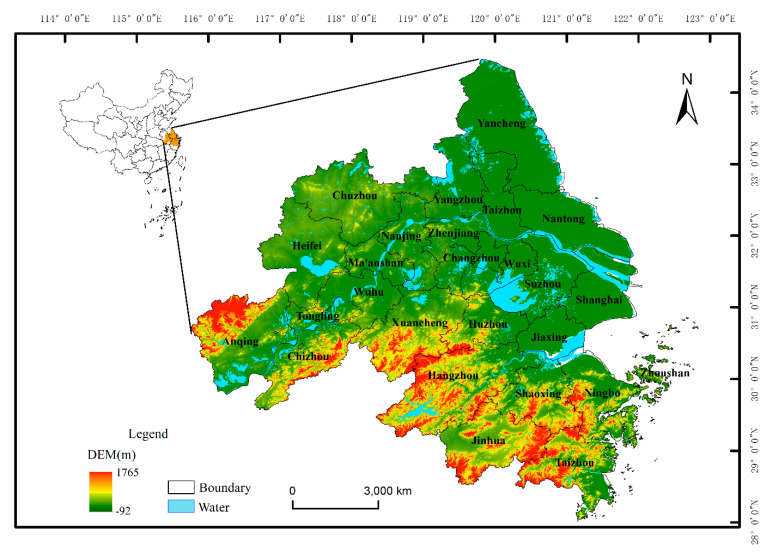
The location of the Yangtze River Delta Urban Agglomeration.

**Figure 2 ijerph-20-01627-f002:**
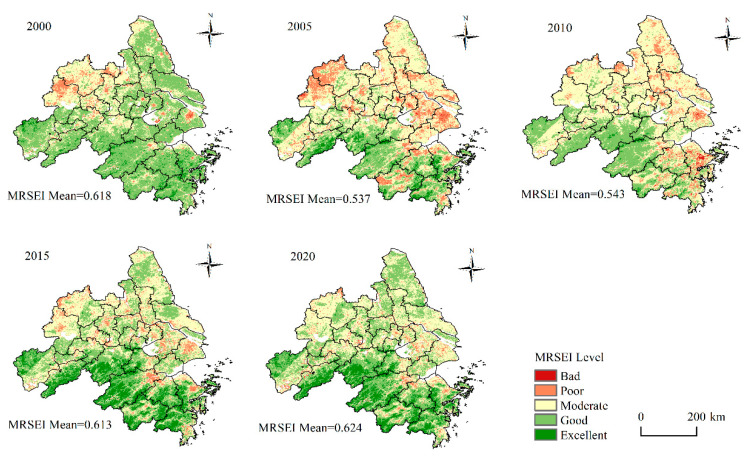
Spatial distribution of EEQ levels in the YRDUA from 2000 to 2020.

**Figure 3 ijerph-20-01627-f003:**
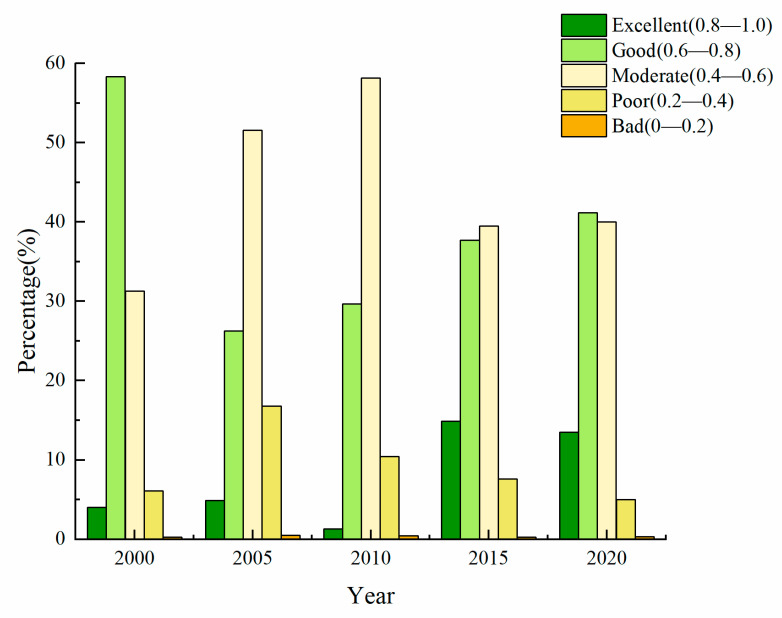
**Distribution** of the EEQ levels in the YRDUA from 2000 to 2020.

**Figure 4 ijerph-20-01627-f004:**
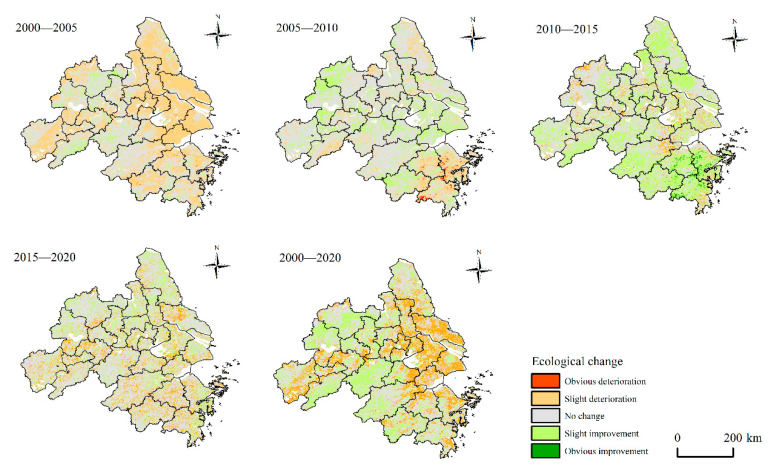
Changes of ecological environment level in the YRDUA from 2000 to 2020.

**Figure 5 ijerph-20-01627-f005:**
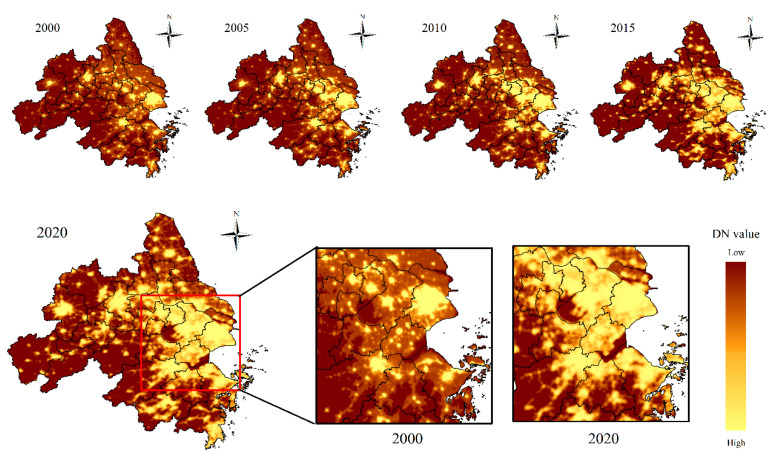
The lighting images and changes of the YRDUA from 2000 to 2020.

**Figure 6 ijerph-20-01627-f006:**
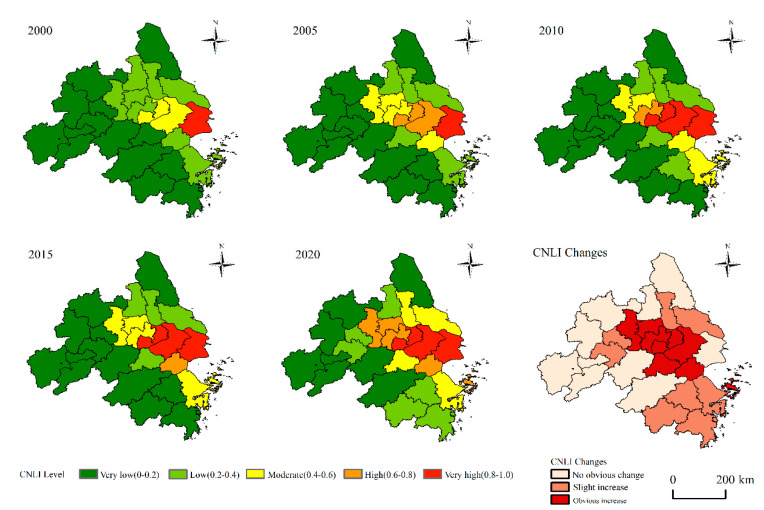
CNLI in prefecture-level cities and its changes.

**Figure 7 ijerph-20-01627-f007:**
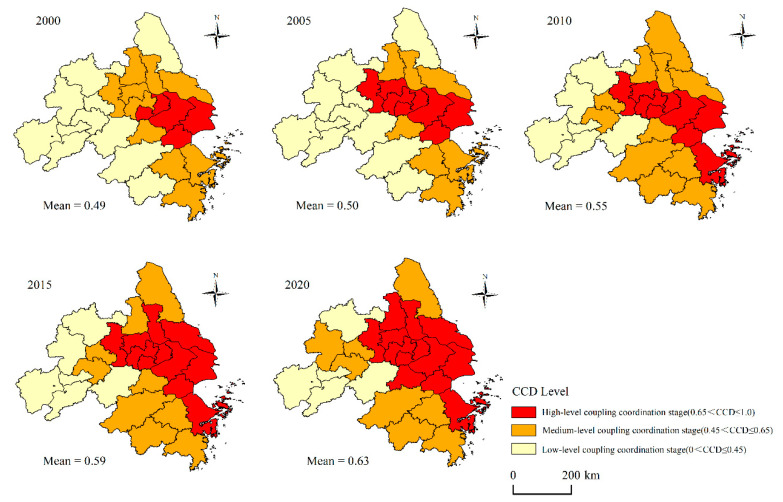
Spatiotemporal changes of CCD in the YRDUA.

**Figure 8 ijerph-20-01627-f008:**
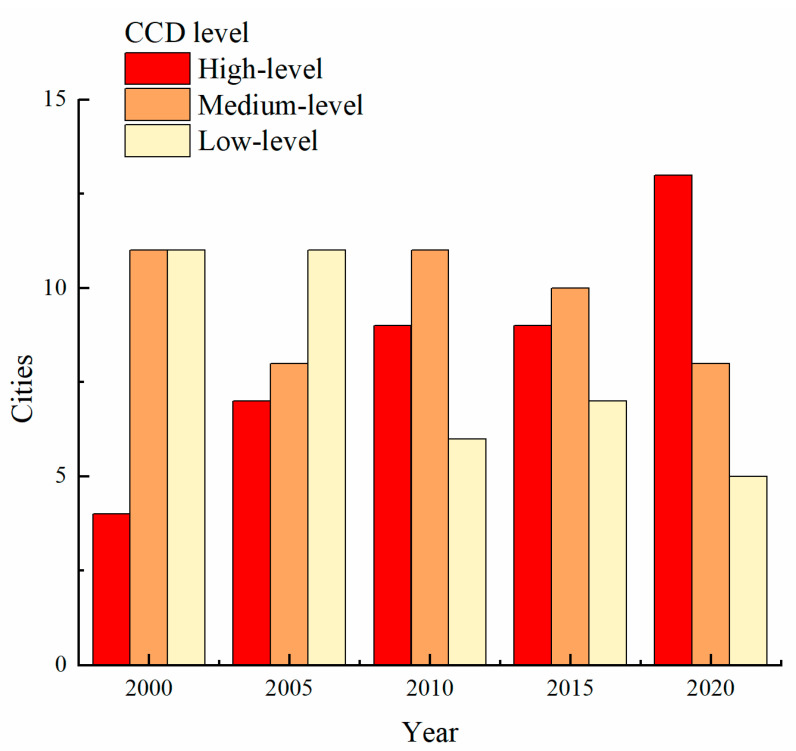
The quantity change of graded CCD in the YRDUA.

**Table 1 ijerph-20-01627-t001:** The data source of five ecological components.

Dataset	Resolution	Time Resolution	Data Description
MOD09A1	500 m	8 Day	MOD09A1 is the surface reflectance data of 1–7 bands of Terra MODIS sensor
MOD11A2	1000 m	8 Day	MOD11A2 is synthesized by daily MOD11A1, and the daytime surface temperature is used in this paper
MOD13A1	500 m	16 Day	MOD13A1 data uses the optimal pixels within 16 days of 500 m resolution, and then calculates the vegetation index of each pixel position
MCD19A2	1000 m	1 Day	MCD19A2 adopts MAIAC algorithm, which can provide accurate and stable aerosol retrieval data

**Table 2 ijerph-20-01627-t002:** Data parameters of DMSP/OLS and NPP/VIIRS.

NTL Data	Sensors	Spatial Resolution	Temporal Resolution	Data Available Interval	Unit
STL	OLS	Annual	30 arc second (around 1 km at equator)	1992—2013	DN (unitless)
VCMSL	VIIRS	Monthly	15 arc second (around 500 m at equator)	April 2012—Present	Nano Watts/cm^2^/Sr

**Table 3 ijerph-20-01627-t003:** Results of the PC1 analysis.

Indictors	2000	2005	2010	2015	2020	Average Value
NDVI	0.8889	0.8761	0.7997	0.9464	0.9551	0.8932
Wet	0.2502	0.1567	0.0848	0.0200	0.0170	0.1057
LST	−0.3709	−0.4388	−0.5873	−0.2304	−0.2869	−0.3829
NDBSI	−0.0741	−0.0046	−0.0034	−0.0072	−0.0242	−0.0227
AOD	−0.0647	−0.1235	−0.0909	−0.2252	−0.0679	−0.1144
Eigenvalue	0.0162	0.0226	0.0276	0.0210	0.0201	0.0215
PC1 Contribution rate (%)	50.25	51.00	54.76	62.26	75.35	58.57

**Table 4 ijerph-20-01627-t004:** Correlation matrix of indexes.

Year	Indictors	NDVI	WET	LST	NDBSI	AOD	*MRSEI*
	NDVI	1.00	0.07	−0.45	−0.51	−0.18	0.84
2000	WET	0.07	1.00	−0.51	−0.74	−0.04	0.32
	LST	−0.45	−0.51	1.00	0.46	0.08	−0.64
	NDBSI	−0.51	−0.74	0.46	1.00	0.07	−0.57
	AOD	−0.18	−0.04	0.08	0.07	1.00	−0.18
	ACC	0.30	0.27	0.30	0.36	0.07	0.51
2005	NDVI	1.00	0.10	−0.36	−0.61	−0.25	0.87
	WET	−0.10	1.00	−0.28	−0.61	−0.12	0.24
	LST	−0.36	−0.28	1.00	0.26	0.18	−0.59
	NDBSI	−0.61	−0.61	0.26	1.00	0.14	−0.59
	AOD	−0.25	−0.12	0.18	0.14	1.00	−0.34
	ACC	0.33	0.28	0.27	0.41	0.17	0.53
2010	NDVI	1.00	0.10	−0.36	−0.61	−0.25	0.87
	WET	0.10	1.00	−0.28	−0.61	−0.12	0.24
	LST	−0.36	−0.28	1.00	0.26	0.18	−0.59
	NDBSI	−0.61	−0.61	0.26	1.00	0.14	−0.59
	AOD	−0.25	−0.12	−0.18	0.14	1.00	−0.31
	ACC	0.33	0.28	0.27	0.41	0.17	0.52
	NDVI	1.00	0.06	−0.52	−0.65	−0.31	0.89
2015	WET	0.06	1.00	−0.35	−0.65	0.27	0.06
	LST	−0.52	−0.35	1.00	0.53	0.10	−0.55
	NDBSI	−0.65	−0.65	0.53	1.00	−0.09	−0.57
	AOD	−0.31	0.27	0.10	−0.09	1.00	−0.39
	ACC	0.39	0.33	0.38	0.48	0.39	0.49
2020	NDVI	1.00	0.03	−0.56	−0.68	−0.30	0.89
	WET	0.03	1.00	−0.32	−0.57	0.14	0.07
	LST	−0.56	−0.32	1.00	0.52	0.19	−0.61
	NDBSI	−0.68	−0.57	0.52	1.00	0.07	−0.63
	AOD	−0.30	0.14	0.19	0.07	1.00	−0.29
	ACC	0.39	0.27	0.40	0.46	0.18	0.50

**Table 5 ijerph-20-01627-t005:** Statistics of changes in *MRSEI* level in the YRDUA from 2000 to 2020.

Year	Change Type	OD		SD		NC	SI		OI	
	Change Level	−4	−3	−2	−1	0	1	2	3	4
2000—2005	Type area/km^2^	128.01		86,664.62		99,559.48	14,676.79		12.03	
	Percent (%)	0.06		43.11		49.52	7.30		0.01	
2005—2010	Type area/km^2^	790.08		29,269.49		130,981.95	40,096.90		6.85	
	Percent (%)	0.39		14.55		65.12	19.93		0.01	
2010—2015	Type area/km^2^	5.77		12,345.76		109,031.59	77,550.12		1651.13	
	Percent (%)	0.01		6.15		54.36	38.66		0.82	
2015—2020	Type area/km^2^	76.34		27,779.07		138,784.97	33,860.49		50.04	
	Percent (%)	0.04		13.85		69.20	16.88		0.03	
2000—2020	Type area/km^2^	111.26		43,565.69		105,987.51	51,310.62		38.04	
	Percent (%)	0.06		21.67		52.73	25.52		0.02	

## Data Availability

The data that support the findings of this study are available from the corresponding author upon reasonable request.

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
