# Peer review of "Analysis of Spatiotemporal Changes of Ecological Environment Quality and Its Coupling Coordination with Urbanization in the Yangtze River Delta Urban Agglomeration, China"

_ijerph, 2023, doi:10.3390/ijerph20021627_

Round 1

Reviewer 1 Report

In my opinion, this article offers a way to obtain data for making decisions related to the management of territorial development. Data-driven is an approach to governance that is increasingly being used in public administration. The article describes an effective way to obtain data for decision-making. The remote sensing technologies are well described and their use, as well as the use of the Modified Remote Sensing Ecological Index, is justified. The authors have taken sufficient care of the comparability of remote images and the minimization of errors. The authors confirmed that MRSEI allows for a quick and effective comparative analysis of regional EEQ and urbanization. All specific objectives of this study have been achieved. However, as a specialist in public administration, I have not found a connection between the results of the study and the social context. For example, there is no explanation of the revealed dynamics of changes in the region. It would be interesting to compare these changes with the policies pursued by the authorities. Can the data obtained help to assess the effectiveness of national or regional policy? Maybe the authors will put this question as a direction for further research, if the focus of the study does not allow them to do it now.

Author Response

Response to Reviewer 1 Comments

We are very grateful to the reviewers for their comments on the manuscript. Due to the limitation of the length of the article, we have not made specific research and analysis on the relationship between the research results and social background. In the next step of research, we will focus on the relationship between the ecological environment change in the study area and the policies implemented by the authorities, and discuss the significance of the research results for future regional policies.

Reviewer 2 Report

This paper calculated Modified Remote Sensing Ecological Index (MRSEI) and the Comprehensive Night Light Index (CNLI), and conducted a quantitative assessment of the YRDUA in China from 2000 to 2020. Then analyzed the spatiotemporal change of MRSEI, eco-environment quality (EEQ), night light, and interaction between EEQ and urbanization. This paper is fully experimental, the results are clear, and the logic is clear. I have the following comments for the authors to consider.

Main comment:

1.     In the introduction, the following works may be helpful for the literature review:

[1]    Xu H, Wang M, Shi T, et al. Prediction of ecological effects of potential population and impervious surface increases using a remote sensing based ecological index (RSEI)[J]. Ecological indicators, 2018, 93: 730-740..

[2]    Yu J, Li X, et al. A remote sensing assessment index for urban ecological livability and its application[J]. Geo-spatial Information Science, 2022: 1-22.

2.     Section 3.1, the MRSEI is calculated by five indexes, however, the weight of the five indexes are not given, which is the core of the MRSEI calculation.

3.    Section 3.1.1-3.1.4, please give the explanation for the equation variables, i.e., ρ1

4.     Section 3.3, it would be better to explain the purpose of the applying of Coupling coordination model, i.e., what does the coupling degree mean, is it used to analyze the relationship between MRSEI and CNLI?

5.     Page 4, Line 152-153, can you explain the sentence detailly, I cannot understand the logical relationship between the “large area and cloud cover” and “use the products to calculate all indicators”.

6.     Page 4, Line 158, “Then the remote 157 sensing image was preprocessed, remove cloud pixels and water mask”. Do you mean the preprocessed remote sensing image is seamless?

7.     The MODIS remote sensing data is impacted by the unwanted noise induced by cloud and other factors, it is unclear whether the authors simply remove the cloud pixels or process them. Since there are many state of art reconstruction methods to solve this issue, it would be worth to be discussed to combine these methods to achieve more reasonable results in the Discuss section.

[1] Li Shuang, Xu Liang, Jing Yinghong, et. al., High-quality vegetation index product generation: A review of NDVI time series reconstruction techniques[J]. International Journal of Applied Earth Observations and Geoinformation, 2021, 105: 102640

[2] Chen, J.M., Deng, F., Chen, M., 2006. Locally adjusted cubic-spline capping for reconstructing seasonal trajectories of a satellite-derived surface parameter. IEEE Trans. Geosci. Remote Sens. 44, 2230–2237

[3] Chu Dong, Shen Huanfeng, et. al., Long time-series NDVI reconstruction in cloud-prone regions via spatio-temporal tensor completion. Remote Sensing of Environment. 264. 2021.

[4] Chen, J., Jonsson, ¨ P., Tamura, M., Gu, Z., Matsushita, B., Eklundh, L., 2004. A simple method for reconstructing a high-quality NDVI time-series data set based on the Savitzky-Golay filter. Remote Sens. Environ. 91, 332–344.

Detailed comment:

1.     Page 2, Line 65, “NDVI, WET, LST and NDBSI”; Page 3, Line 107-108,” DMSP/OLS, NPP/VIIRS”. Abbreviations that appear for the first time should be provided with the full English name.

2.     Page 3, Line 120, there's an extra space in “EEQ of  the YRDUA”.

3.     There are some abbreviations whose full English name appeared twice. Page 4, Line 150, “MODIS”; Page 5, Line 197, “MRSEI”; Page 7, Line 245, ”CNLI”;

4.     Page 5, Line 180, why the abbreviation of “Pseudo Invariant Target Calibration Model” is PBPIF?

5.     Page 5, Line 204, the full English name of NDVI is “normalized differential vegetation index.

6.     Page 6, Formula 2-7, the position of formula label is not uniform.

7.     Page 17, Line 544, should the “of” in “south was better than that of the north” be changed to “in”?

8.     Fig. 8, CDC level? 

Reviewer 3 Report

Presented manuscript concentrated on analysis of spatiotemporal changes of ecological environment quality (EEQ) and its coupling coordination with urbanization in the Yangtze River Delta Urban Agglomeration (YRDUA), China. Based on the Modified Remote Sensing Ecological Index (MRSEI) and the Comprehensive Night Light Index (CNLI), the Author/Authors conducted a quantitative assessment of the impact of the urbanization process on the ecological environmental quality (EEQ) in YRDUA in China from 2000 to 2020.

The topic is interesting and actual. The main ideas of the article and the questions posed and analyzed by the Author/Authors are relevant for the International Journal of Environmental Research and Public Health.

I appreciate the article, the proposed research methodology, data and tools used can be used by different economic and planning sectors at different levels, in all areas where knowledge of the impact of the urbanization process on the ecological quality of the environmental is valuable.

After reading the article, the following comments came to mind:

(1) The Introduction  section (line 117-129) should be strengthened to clearly summarise what new has been found, in order to demonstrate the contribution of this study to the development of this topic.

(2) Fig 1; Please improve subtitle visibility of the map.

(3) 2.2.1. MODIS Data section; Please provide the source of this remote sensing data.

(4) 2.2.2. Nighttime remote sensing data section;  Additionally I suggest making a table listing  the nighttime remote sensing datasets, their sources and characteristics, their scale of compilation, their validity date, as a summary… similar to Tab. 1

(5) Research Methods section; The methodology is well developed but please complete the information on the all designations used in the equation 2, …where r2 is…. and r1 is….  and in other equations 3 – 10 …. Not every reader, researcher knows the spectral characteristics and bands of satellite imagery, Additional explanation of the designations used in the formulas  would increase the value of the article…  Line 241-242; this is very general information.

(6) Line 275 -276; It is only in section 4. Results we learn about the first principal component analysis used, the methodology section should be supplemented with these issues..

(7) Line 276 „…gathering most of the information from…”; It's debatable…,  first components explained slightly more than 50% of the overall variability all indicators of MRSEI (2000 – 2015).

(8) Line 279; The Authors write about the use of correlation coefficient (Spearman’s). Please also complete in this methodology section the information on the accepted significance level (p<α = ???), against which the probability of the test statistic  was verified, whether the correlation coefficients analysed was statistically significant.

(9) Line 284-285 „MRSEI is still the largest, higher than NDVI, WET, 284 LST, NDBSI, and AOD at 0.51, 0.18, 0.22, 0.19, 0.09, and 0.31, respectively”; What the authors have in mind..?

(10) Line 294-296 „Results show that the MRSEI of the YRDUA fluctuated and increased from 2000 to 2020, with the lowest mean of only 0.537 in 2005 and increased to 0.624 in 2020” ; In what unit are the MRSEI values? From equation 1 we learn that MRSEI is a function of NDVI, WET, LST, NDBSI, and AOD, How did the Authors calculate the MRSEI? as the sum of the standardized indexes?

(11) Line 312-313; What could have caused this, what kind of processes?

(12) Line 401-402 and fig. 6; How the ranges were established of CNLI (0-0.2, 0.2-0.4, and so on..)

(13) Line 468, Reviewing the past 2 years; or 20 years?

(14) No reference in the text to Figure 8 …

(15) Line 497-499; Have the Authors made an evaluation, validation of this model ?

(16) Line 506-508; I agree with the Authors..

(17) I  suggest reviewing a little bit the Discussion, Chapter 5 is not a scientific discussion! This chapter is of a concluding nature and includes assumptions for future research. There is a lack of reference of the results obtained to other researchers, which is important in scientific publication!

(18) With reference to the text read, further questions arises...

For other regions of China and the world, would the proposed indexes to be modified due to the specificities of the region, the country...?

In which areas, geographical regions do the Authors intend to conduct empirical research, or only in the Yangtze River Delta Urban Agglomeration?

Round 2

Reviewer 2 Report

The authors have addressed most of my concerns. The last one issue is that the authors discussed the importance of the image denoising in Page 18 Line 562-566, but some references are needed. 

[1] Li Shuang, Xu Liang, Jing Yinghong, et. al., High-quality vegetation index product generation: A review of NDVI time series reconstruction techniques[J]. International Journal of Applied Earth Observations and Geoinformation, 2021, 105: 102640

[2] Chen, J.M., Deng, F., Chen, M., 2006. Locally adjusted cubic-spline capping for reconstructing seasonal trajectories of a satellite-derived surface parameter. IEEE Trans. Geosci. Remote Sens. 44, 2230–2237

[3] Chu Dong, Shen Huanfeng, et. al., Long time-series NDVI reconstruction in cloud-prone regions via spatio-temporal tensor completion. Remote Sensing of Environment. 264. 2021.

[4] Chen, J., Jonsson, ¨ P., Tamura, M., Gu, Z., Matsushita, B., Eklundh, L., 2004. A simple method for reconstructing a high-quality NDVI time-series data set based on the Savitzky-Golay filter. Remote Sens. Environ. 91, 332–344.

Author Response

We have added the corresponding reference (reference [61-64]) when we discussed the importance of image denoising on page 18, lines 562-566.